# Precision Oncology in Gastrointestinal Stromal Tumors

Hiba Mechahougui [1], Montemurro Michael [2] and Alex Friedlaender [1,3,*]

1    Oncology Department, Geneva University Hospital, 1205 Geneva, Switzerland; hiba.mechahougui@hcuge.ch
2    Genolier Cancer Center, 1272 Genolier, Switzerland
3    Clinique Générale Beaulieu, 1206 Geneva, Switzerland
*    Correspondence: afriedlaender@beaulieu.ch

**Abstract:** GIST (gastrointestinal stromal tumors) represent 20% of sarcomatous tumors and 1–2% of primary gastrointestinal cancers. They have an excellent prognosis when localized and resectable, though their prognosis is poor in the metastatic setting, with limited options after the second line until recently. Four lines are now standard in KIT-mutated GIST and one in PDGFRA-mutated GIST. An exponential growth of new treatments is expected in this era of molecular diagnostic techniques and systematic sequencing. Currently, the main challenge remains the emergence of resistance linked to secondary mutations caused by selective pressure induced by TKIs. Repeating biopsies to tailor treatments might be a step in the right direction, and liquid biopsies at progression may offer a non-invasive alternative. New molecules with wider KIT inhibition are under investigation and could change the catalog and the sequence of existing treatments. Combination therapies may also be an approach to overcome current resistance mechanisms. Here, we review the current epidemiology and biology of GIST and discuss future management options, with an emphasis on genome-oriented therapies.

**Keywords:** gastrointestinal stromal tumor; metastatic GIST; precision oncology; tyrosine kinase inhibitors; imatinib; KIT; PDGFR; regorafenib; sunitinib

## 1. Introduction

### 1.1. Epidemiology

GIST (gastrointestinal stromal tumors) represent 20% of sarcomatous tumors and 1–2% of primary gastrointestinal cancers. They typically present in older individuals and are most common in the stomach (60–70%), followed by the small intestine (20–25%), colon and rectum (5%), and esophagus (<5%) [1]. GIST can also occur in children and young adults, especially when a genetic predisposition is involved [2]. They are equally common in male and female patients [3].

### 1.2. Clinical Presentation

GIST are often asymptomatic and an incidental finding on imaging. However, the clinical presentation can include abdominal pain or digestive bleeding. Endoscopic and radiologic findings are variable and often show a rather homogeneous lesion if the tumor is small, but if larger, one with heterogeneous enhancement, irregular margins, central necrosis, and signs of hemorrhage [4,5].

### 1.3. Histological Diagnosis

GIST cells derive from Cajal's interstitial cells, or its precursors, and constitutively express the KIT protein/receptor. GIST occur nearly always in the gastrointestinal tract. While many of the very heterogeneous soft-tissue tumors are difficult to diagnose, GIST are well defined by a combination of morphologic, immunohistochemical, and molecular features.

Histologically, there are two main types, spindle-cell GIST, mainly found in mutated *KIT* or *BRAF* GIST (in 70% of cases), and epithelioid-cell GIST (20%), mainly found in platelet-derived growth factor receptor A *(PDGFRA)* or succinate dehydrogenase *(SDH)* GIST. Ten percent harbor a mixed morphology [6,7].

### 1.4. Immunohistochemistry

DOG-1 (discovered on GIST-1) has both high sensitivity and specificity and is found in 88% of cases [8]. *KIT* is found in 95% of GIST and is very sensitive but not specific. However, in GIST with a *PDGFRA* mutation, the sensitivity of these markers decreases to 9% and 79%, respectively [9]. These two markers are classically expressed in a diffuse and intense manner.

### 1.5. Molecular Pathogenesis

*KIT* is a proto-oncogene that encodes the KIT tyrosine kinase (TK) receptor. It includes two main regions. The first is the receptor regulatory domain with the extracellular region, the transmembrane region, and the juxta-membrane domain. The second is the cytoplasmic region with a TK domain that includes a TK1 domain that anchors ATP (encoded by exons 13 and 14) and a TK2 domain that binds and phosphorylates downstream substrates. An activation loop (encoded by exon 17) is located on the TK2 domain and stabilizes the activated KIT receptor in a permanently active state. Binding of the ligand to KIT results in the activation of tyrosine kinase activity and stimulation of downstream pathways, including RAS/RAF/MAPK, PI3K/Akt/mTOR, and Src kinase pathways, resulting in cellular proliferation and inhibition of apoptosis. Activating mutations in KIT leads to constitutive activation of KIT in a ligand-independent manner. PDGFRA is structurally similar [7,10] (Figure 1).

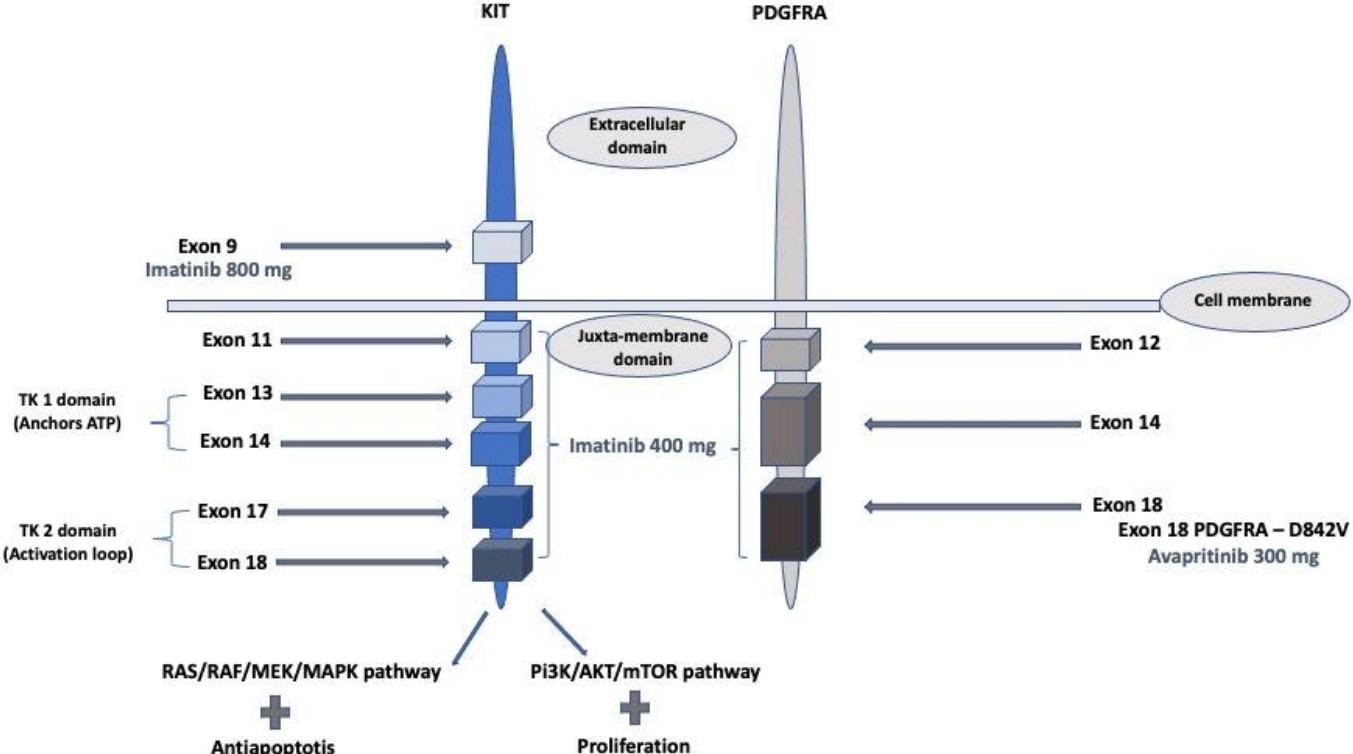

**Figure 1.** KIT and PDGFRA gain-of-function mutations and related treatment with recommended dosage (imatinib and avapritinib).

The majority of these gain-of-function mutations are found in the KIT gene, with two-thirds and the most frequent within exon 11. Exon 9 mutations are often found in small or large bowel tumors and represent approximately 10% of GIST. Deletions, deletion–insertion, point mutations, duplications, insertions, and inversion have been identified in *KIT*. Deletions in exon 11, especially involving codons 557 and 558, are associated with a poorer natural prognosis compared to exon 11 point mutations [11]. Mutations in exons 13, 17, and 18 are rare [12].

Furthermore, 5 to 10% of mutations are related to the *PDGFRA* oncogene, more specifically to exons 12, 14, and 18. *PDGFRA* mutations are more commonly found in the stomach. Exon 18 of *PDGFRA* is the most frequently mutated region, with exon 18 D842V mutations accounting for 70% of *PDGFRA*-mutant cases, while exons 12 or 14 are rarely mutated. [12] KIT and PDGFRA mutations are considered mutually exclusive [13].

Despite extensive sequencing, some GIST remain "wild-type" GIST, but a number have been identified nowadays with the modern techniques to carry low-frequency KIT or PDGFRA mutations, while others were associated with fusions of NTRK (NTRK3-ETV6) and FGFR1 (FGFR1-HOOK3, FGFR1-TACC1), alterations in the RAS-MAPK pathway with *BRAF* mutations, *NF1* mutations, or SDHA deficiency caused by a germline mutation in the suppressor genes encoding the SDH complex or by SDHC promotor methylation [13,14].

## 2. Management of Localized GIST

Surgery is the treatment of choice for resectable GIST if no major functional losses are to be expected. The tumor should be resected completely, and rupture needs to be avoided [5]. Patients at high risk of recurrence need post-operative imatinib for a duration of a minimum of 3 years if well tolerated [15].

Different scores are used to determine the risk:

The "Miettinen classification", also known as the Armed Forces Institute of Pathology (AFIP) classification, includes the mitotic index, the size of the tumor, and the location.

Joensuu's classification, also known as modified NIH classification, uses the parameters above and integrates the pejorative nature of a perforation. It aims to better split the GIST between intermediate and high risk [16].

All classifications stratify patients into very low, low, intermediate, and high-risk categories of recurrence. The indication for adjuvant treatment depends on the risk score, but also on the mutational status [17]. As an example, patients with the PDGFRA D842V mutation do not receive adjuvant treatment as this is a resistance mutation to imatinib [18].

Molecular prognostic factors such as the level of tumor genome rearrangement have been investigated and are currently evaluated in intermediate-risk GIST in the GI-GIST trial (NCT02576080).

## 3. Management of Metastatic GIST

### 3.1. First-Line Treatment in the Metastatic Setting: Imatinib

Imatinib is an inhibitor of KIT, PDGFRA, and BCR-ABL tyrosine kinase and is used to treat inoperable or metastatic GIST. Standard dosage is 400 mg for all sensitive mutations. Indeed, sensitivity to imatinib for exon 9 and exon 11 mutated GIST was demonstrated in the lead trial evaluating imatinib in GIST [19], while KIT exon 9 mutations are treated with 800 mg/d as this provides longer progression-free survival (PFS) [20–22].

Currently, imatinib is continued until progression or intolerance in the metastatic setting.

The exon 13 and 17 activation loop mutations are essentially secondary mutations that occur on imatinib therapy. Primary exon 13 mutations exist and are often sensitive to imatinib in vitro. Given the rarity of this mutation, in vivo sensitivity is not clear as little evidence exists to date [23].

Currently, imatinib is continued until progression or intolerance in the metastatic setting.

Primary PDGFRA mutations occur mainly in exon 18 and exon 12, which, respectively, encode the activation loop in the juxta-membrane domain, but more rarely in exon 14, which encodes the ATP-binding domain [24]. While D842V, the most common exon 18 mutation, confers a primary mutation to imatinib, other types of PDGFRA mutations are sensitive to imatinib [25].

### 3.2. Primary Resistance to Treatment

#### 3.2.1. Pseudo-Resistance: Imatinib Plasma-Levels and Pharmacokinetics

Demetri et al. showed in a small group of patients that imatinib plasma levels above 1100 ng/mL were associated with clinical benefit and a longer time to disease progression [26]. Pointing in the same direction, in chronic myeloid leukemia, Gotta et al. proved in a prospective randomized controlled trial that imatinib dose monitoring helps in achieving efficient plasma concentrations [27]. In another trial, durable effective imatinib concentrations were reached only by 33.3% [28], raising the question if imatinib should be dosed individually.

Several treatments such as proton pump inhibitors can have interactions with oral oncology treatments. Indeed, the concomitant use of TKIs and proton pump inhibitors can reduce TKI absorption, thus potentially reducing the effectiveness of TKIs [29].

Genetic polymorphisms of cytochrome P450 can participate in interpatient variability in imatinib blood levels [30]. At progression, the use of high-dose imatinib (800 mg daily) has shown benefit to patients with advanced or metastatic GIST that progressed on the standard dose [31].

Compliance issues are also a major factor to consider when suspecting pseudo-progression. Indeed, in a study made on imatinib-treated patients who had a diagnosis of CML or GIST, compliance with imatinib was about 75%, with 30% of patients interrupting therapy for at least 30 consecutive days in the first year [32].

#### 3.2.2. PDGFRA Exon 18 D842V Mutation (75% of PDGFRA Mutations)

Most *PDGFRA* mutations are sensitive to imatinib, with the exception of the frequent PDGFRA exon 18 D842V mutation, which confers primary resistance to imatinib [33].

Avapritinib has been specifically developed to target this mutation and the NAVIGATOR study showed high efficacy of this drug with above 90% overall response rate (ORR) and a duration of response of 70% at 1 year [34–37].

To note, among 167 patients starting on 300 mg of avapritinib, 37.0% of all patients and 52.0% of patients older than 65 years showed cognitive dysfunction. This toxicity decreased faster to a lower grade with dose modification (1.3–3.1 weeks) than without (4.9–7.6 weeks). Median PFS was 11.4 months with dose modification and 7.2 months without. As a result, early recognition of neuro-toxicity and adapted dose modification can help maintain patients on this treatment. It can be recommended to assess cognitive function at baseline and monitored [38].

#### 3.2.3. Rare Non-KIT/PDGFRA Molecular Subtypes

In the past, GIST have been classified as KIT or PDGFRA mutated according to testing that included the frequent mutations. GIST missing these mutations have been classified as "wild type" (WT). Later, the concept of quadruple negative WT-GIST was coined, and additional subgroups were identified. Recently, new techniques have detected low-frequency mutations [39]. The latter probably explains why some GIST considered to be WT respond to TKI [40,41].

Within the WT-GIST category, the following "new" subtypes have been identified.

SDH-Deficient (5 to 8%) [42]

About 20–40% of WT-GIST show defects in succinate dehydrogenase (SDH) complexes. Some mutations can be germline [43] and involve genetic testing. Few data are available in *SDH*-deficient GIST due to their rarity. The early, large trials did not classify WT-GIST further [44].

A retrospective analysis of 87 patients with SDH-deficient GIST found better response rates with sunitinib than with imatinib, which had very little activity in this cohort. Nilotinib and vandetanib have also been used, but limited data do not support the use of these drugs [45].

The dysfunction of the SDH complex in these tumors leads to a pseudo-hypoxic phenotype suggesting a potential benefit of TKIs with anti-angiogenic activity. In the phase II REGISTRI trial, an ORR of nearly 20% was achieved with sunitinib, and regorafenib was associated with an 87% disease control rate [46]. Other TKIs such as regorafenib and pazopanib showed limited efficacy.

A novel third-generation TKI, olverembatinib, has shown antitumor activity in patients with TKI-resistant SDH-deficient GIST in a phase Ib/II study with 2 partial responses in 6 evaluable patients and 1 with stable disease for 36 cycles, calling for its further investigation [47].

As SDH-loss causes succinate accumulation and activation of pseudohypoxia signaling [48] via overexpression of HIF-proteins, specific HIFa inhibitors such as belzutifan are under development in different types of tumors associated with overexpression of HIF (NCT04895748, NCT04924075) and could be good candidates for SDH-deficient GIST in the future.

Furthermore, temozolomide has potential interest in this subgroup with promising results in five patients (100% disease control rate, 40% ORR), motivating an ongoing phase II study (NCT03556384). Overexpression of FGF/FGFR signaling pathways has also been reported in SDH-deficient GIST, and a new pan-FGFR inhibitor, rogaratinib, is being evaluated in a phase II trial (NCT04595747).

Non-SDH Deficient

NTRK fusions

NTRK fusions are agnostic molecular alterations that render tumors sensitive to TRK inhibitors. Larotrectinib or entrectinib demonstrated activity against solid tumors harboring NTRK fusions [49–52] and can be used in this setting.

BRAF V600E mutations: (0.6 to 3.9%)

Just like NTRK fusions, *BRAF V600E* mutations are considered tumor-agnostic features predictive to response to BRAF inhibitors. In a study of GIST, BRAF V600E mutations were detected in two of 28 KIT and PDGFRA wild-type patients. [53]. BRAF mutations confer resistance to imatinib and sunitinib [54].

NF1 mutations (0.1 to 2.4%)

Neurofibromatosis 1 (NF1) is an autosomal dominant disorder caused by germline mutations in the *NF1* tumor suppressor gene causing its inactivation. There is an estimated 7% of individuals with NF1 who will develop GIST during their life [42].

To date, there is no standard treatment for NF1-mutated GIST, which do not respond to TKIs, and surgery remains the main option for these patients. However, these tumors appear to be more indolent [55].

MEK inhibitors might have clinical efficacy in other NF1-associated tumors, given the role of the RAF/MEK/ERK pathway in neurofibromas. A phase II trial with mirdametinib, a MEK inhibitor, for adolescents and young adults with NF1-associated plexiform neurofibromas showed a 42% partial response and a significant decrease in pain ratings [56].

*3.3. Secondary Resistance*

3.3.1. KIT and PDGFRA Secondary Mutations

In the second line, 67% of patients have one or more secondary mutations involving KIT exon 17, exon 13, and exon 14, causing resistance to imatinib [57].

Point mutations associated with imatinib resistance are usually located in the drug/ATP-binding pocket of the receptor (encoded by exons 13 and 14) or in the activation loop (encoded by exon 17) [58].

For avapritinib, mechanisms of secondary resistance in PDGFRA-mutant GIST involve compound mutations of exons 13, 14, and 15 of *PDGFRA* with codon 658 and 680 mutations representing a recurrent cause of resistance [59].

3.3.2. Second-Line Sunitinib

Sunitinib was approved after a phase III trial in patients with GIST failing or intolerant to imatinib and showed a longer time to progression (27.3 versus 6.4 weeks, $p < 0.0001$) in patients with sunitinib than placebo.

Of interest, higher response rates were observed among GIST with a primary KIT exon 9 mutation [60].

Antiangiogenic effects of sunitinib treatment may contribute to its effectiveness. Indeed, sunitinib selectively inhibits PDGFRB and VEGFR in addition to KIT and PDGFRA, whereas imatinib inhibits PDGFRB but not VEGFR. Sunitinib has, however, little activity against secondary mutations involving the KIT activation loop (exons 17 and 18) [61].

3.3.3. Third-Line Regorafenib

Regorafenib is a multikinase inhibitor and also inhibits VEGFR and showed a significant improvement in median PFS (4.8 vs. 0.9 months, HR 0.27, $p < 0\cdot0001$) over placebo in patients already treated with imatinib and sunitinib, leading to its approval in the third line [62]. It has become the treatment of choice in patients with an exon 17 mutation, as these do not respond to sunitinib [63].

3.3.4. Third-Line Pazopanib

Pazopanib has been tested in the third line, following imatinib and sunitinib failure, and showed modest benefit with a median PFS of 3.4 months (95% CI 2.4–5.6) vs. 2.3 months in best supportive care only [64].

3.3.5. Fourth-Line Ripretinib

Ripretinib is an anti-KIT anti-PDGFRA TKI, also active against PDGFRB, TIE2, VEGFR2, and BRAF. The INVICTUS trial enrolled 129 participants with advanced GIST who progressed after imatinib, sunitinib, and regorafenib. Ripretinib improved PFS and OS significantly (PFS 6.3 vs. 1 month, HR 0.15, $p < 0.0001$, OS 15.1 months vs. 6.6 months HR 0.36, $p = 0.0004$) [65]. A further gain in PFS of 3.7 months was obtained by doubling the dose ($2 \times 150$ mg) upon first progression on the standard dose [66].

In this setting, ripretinib demonstrated a PFS benefit regardless of the primary mutation [67].

Ripretinib did not improve PFS when tested in the second-line INTRIGUE trial against sunitinib, but improved response rates (23.9 vs. 14.6%). OS data are immature. Sunitinib showed better PFS in the exon 9 subgroup [68].

The failure of ripretinib in the second line compared to sunitinib might be explained by the emergence of secondary exon 13 mutations, which decrease ripretinib efficacy compared to sunitinib [68].

Interestingly, a Chinese phase II study, testing ripretinib in the second line, showed a benefit in PFS for ripretinib [69].

### 3.3.6. Third-Line Avapritinib

VOYAGER, a randomized, phase III trial, tested avapritinib versus regorafenib in the third line. In molecularly unselected patients, the primary end point was not met. There was no significant difference in median PFS between avapritinib and regorafenib, but in the selected subgroup of patients with the PDGFRA exon 18 D842V mutation, which is resistant to the other TKIs, avapritinib showed high response rates [70] (Table 1).

**Table 1.** Gain-of-function mutations and related treatment with recommended dosage.

| KIT | First-Line Treatment | Suggestion of Second Line | Suggestion of Third Line | Suggestion of Fourth Line |
|---|---|---|---|---|
| Exon 9 | Imatinib 800 mg | Sunitinib ** | Regorafenib *** | Ripretinib 150 mg |
| Exon 11 | Imatinib 400 mg | Sunitinib | Regorafenib | Ripretinib 150 mg |
| Exon 13 | Imatinib 400 mg, except V654A mutation | | Regorafenib | Ripretinib 150 mg |
| Exon 17 | Regorafenib *[1] | Ripretinib 150 mg | Avapritinib 300 mg | Consider enrollment in a study protocol |
| **PDGFRA** | | | | |
| Exon 12 | Imatinib 400 mg | Sunitinib | Regorafenib | Ripretinib 150 mg |
| Exon 14 | Imatinib 400 mg | Sunitinib | Regorafenib | Ripretinib 150 mg |
| Exon 18 (except D842V) | Imatinib 400 mg | Sunitinib | Regorafenib | Ripretinib 150 mg |
| Exon 18 D842V | Avapritinib 300 mg | Ripretinib 150 mg | Trial/Study? | Trial/Study |
| **SDH deficient** | Sunitinib? Regorafenib? | Temozolomide? | Consider enrollment in a study protocol | Consider enrollment in a study protocol |
| **NTRK** | TRK inhibitors | 2nd generation TRK inhibitors | Consider enrollment in a study protocol | Consider enrollment in a study protocol |
| **BRAF V600E** | BRAF inhibitors | Consider enrollment in a study protocol | Consider enrollment in a study protocol | Consider enrollment in a study protocol |
| **NF1** | Consider enrollment in a study protocol in case of symptomatic progression | Consider enrollment in a study protocol | Consider enrollment in a study protocol | Consider enrollment in a study protocol |

*[1] Primary mutations in exon 17 are extremely rare and are mainly found in the case of secondary mutations after first-line imatinib. ** Sunitinib standard dosing is 50 mg/d for three weeks, then a 2 week break. Most often 37.5 mg/d until progression or patients' preference. *** Regorafenib standard dosing is 160 mg/d for 3 weeks out of 4, but alternative schedules are also used (see text). *PDGFRA*: platelet-derived growth factor receptor A; *SDH*: succinate dehydrogenase; *NTRAK*: neurotrophic tyrosine receptor kinase, *BRAF*: v-raf murine sarcoma viral oncogene homolog B1; *NF1*: neurofibromin 1.

## 4. Future Perspectives (Table 2)

### 4.1. Immunotherapy

The activity of anti-PD-1/PD-L1 and CTLA-4 drugs and combinations have been investigated in GIST. In a randomized phase II trial with 40 patients, nivolumab with or without ipilimumab showed only modest response rates. Other clinical trials based on the same combination are ongoing [71]. No or little clinical efficacy was seen with ipilimumab plus dasatinib [72], epacadostat with pembrolizumab [73], and in the subgroup of 31 patients with GIST of the PEMBROSARC trial treated with pembrolizumab and metronomic cyclophosphamide, which showed a low 2.3% ORR [74].

**Table 2.** Ongoing studies evaluating drugs for metastatic or advanced unresectable GIST after standard treatment, and first line for SDH mutant/deficient.

| Trials | Phase | Drug | Control Arm | Population | Status |
|---|---|---|---|---|---|
| | | | TKIs | | |
| NCT05489237 | I | IDRX-42 | | Metastatic and/or unresectable GIST. After at least 1 line including imatinib | Recruiting |
| NCT03594422 | I | Olverembatinib | | Metastatic and/or unresectable GIST with primary resistance to imatinib. | Recruiting |
| NCT05160168 | I/II | THE 630 | | Metastatic and/or unresectable GIST. After at least 1 line including imatinib | Recruiting |
| NCT04595747 | II | Rogaratinib | | Metastatic and/or unresectable SDH-deficient GIST. No prior treatment required. | Recruiting |
| NCT04193553 (LENVAGIST) | II | Lenvatinib | | Metastatic and/or unresectable GIST. After at least 2 lines including imatinib, and sunitinib | Recruiting |
| NCT04409223 | III | Famitinib | Sunitnib | Metastatic and/or unresectable GIST. Second line after imatinib failure | Recruiting |
| NCT05208047 | III | 2 experimental arms: ⇨ Bezuclastinib plus sunitinib ⇨ Bezuclastinib | Sunitinib | Metastatic and/or unresectable GIST. After at least 1 line including imatinib | Recruiting |
| | | | Immunotherapy | | |
| NCT05152472 (ATEZOGIST) | II | Atezolizumab + imatinib | Imatinib | Metastatic and/or unresectable GIST. After at least 3 lines including imatinib, sunitinib and regorafenib | Recruiting |
| NCT03609424 | Ib/II | Spartalizumab (PDR001) + imatinib | | Metastatic and/or unresectable GIST. After at least 3 lines including imatinib, sunitinib and regorafenib | Completed |
| NCT04000529 | Ib | TNO155 + Spartalizumab | | Metastatic and/or unresectable solid tumor, after failure of standard therapies. | Recruiting |
| NCT03475953 (REGOMUNE) | I/II | Regorafenib + avelumab | | Metastatic and/or unresectable solid tumor, after failure of standard of care | Recruiting |
| NCT04258956 (AXAGIST) | II | Axitinib + avelumab | | Metastatic and/or unresectable GIST. After at least 2 lines including imatinib, and sunitinib | Recruiting |
| NCT02834013 | II | Ipilimumab + nivolumab | | Metastatic and/or unresectable solid tumor, after failure of standard therapies. | Recruiting |
| | | | Chemotherapy | | |
| NCT03556384 | II | Temozolomide | | Metastatic and/or unresectable SDH-deficient GIST. No prior treatment required | Recruiting |
| NCT03944304 | II | Paclitaxel | | Metastatic and/or unresectable GIST. After at least 3 lines including imatinib, sunitinib and regorafenib | Recruiting |

**Table 2.** *Cont.*

| Trials | Phase | Drug | Control Arm | Population | Status |
|---|---|---|---|---|---|
| Other | | | | | |
| NCT05245968 (CHAPTERGIST-101) | I | Pimitespib (TAS-116) in Combination with Imatinib | | Metastatic and/or unresectable GIST. After imatinib. | Recruiting |
| NCT04006769 | I | Entacapone + imatinib | | Metastatic and/or unresectable GIST. After at least 2 lines including imatinib, and sunitinib | Active, not recruiting |
| NCT03411915 | I | Tidutamab | | Metastatic and/or unresectable GIST. After at least 2 lines including imatinib, and sunitinib | Completed |

Ongoing trials are investigating avelumab with axitinib (NCT04258956, AXAGIST) or with regorafenib (NCT0347595, REGOMUNE), spartalizumab with imatinib (NCT03609424), or with TNO155 or ribociclib (NCT04000529).

The retrospective analysis of the Sarc028 study showed the presence of tertiary lymphoid structures (TLSs), found to be associated with better response and longer PFS [75]. Today, immunotherapy in GIST requires further evaluation in prospective trials, possibly guided by new biomarkers.

### 4.2. New Tyrosine Kinase Inhibitors

Lenvatinib, a broad-spectrum TKI targeting KIT, RET, PDGFRA, VEGFR 1-3, and FGFR 1-4, is investigated in the third line in LENVAGIST, a phase II study (NCT04193553).

Bezuclastinib (CGT9486) showed good clinical benefit and a median PFS of 12 months in an early phase trial and is currently tested with or without sunitinib in a phase 3 clinical trial (NCT05208047).

THE 630, a pan KIT inhibitor is in testing in a phase I/II study and showed good preclinical results (NCT05160168).

### 4.3. Pimitespib

Heat shock protein 90 is necessary for the stabilization of KIT and PDGFRA. Several HSP90 inhibitors had preclinical activity in GIST.

In a randomized placebo-controlled phase III trial (CHAPTERGIST-301), pimitespib, also known as TAS 116, a novel HSP90 inhibitor tested in the fourth line, showed an improved PFS and OS compared with placebo in patients with previously treated advanced GIST. Exploratory pharmacogenomic analysis showed a benefit irrespective of KIT mutation status. As a result, this therapy has been approved in the fourth line in Japan for the treatment of metastatic GIST. Visual impairment was reported in patients receiving pimitespib, with 13% of night blindness, and two cases of retinal vein occlusion and visual impairment, which resolved with discontinuation [76].

A phase I study evaluating pimitespib in combination with imatinib (NCT05245968, CHAPTERGIST-101) is ongoing.

### 4.4. Intratumoral Vaccination

Ilixadencel is a cell-based immune primer injected intratumorally that has been clinically investigated in metastatic renal cell carcinoma and hepatocellular carcinoma. It has been evaluated in a phase I study and presented an acceptable safety profile and radiological tumor responses in 33% of treated patients. Further investigation is needed [77].

## 5. Discussion

Despite the rarity of GIST, multicentric and multinational trials have led to the approval of four lines of treatment, and an exponential growth of new treatments is expected in this era of molecular diagnostic techniques and systematic sequencing.

Currently, the main challenge remains the emergence of resistance linked to secondary mutations caused by selection pressure induced by TKIs [78]. Future studies might select patients according to the secondary mutations, either on the basis of a (repeated) solid or liquid biopsy. Indeed, the failure of ripretinib in the second line compared to sunitinib might be explained by the emergence of secondary exon 13 mutations, which decrease ripretinib efficacy compared to sunitinib [62]. Furthermore, liquid biopsies may overcome the challenge of tumor heterogeneity and aid in detecting relevant resistance clones. An exploratory study of the NAVIGATOR trial for PDGFRA-mutant GIST showed that mutant ctDNA was detected in 63% of patients, and in this same population, the median sum of target lesions was 18.2 cm, suggesting that ctDNA detection might be potentially limited by the tumor burden [59]. Further research is needed to validate the optimal approach [79–82].

Imatinib has been the first-line standard since the discovery of its efficacy in GIST, and current studies focus mainly on later lines given the excellent tolerance profile of the molecule. New studies might focus on earlier lines to delay the emergence of resistance. Nonetheless, imatinib remains the gold standard, and the use of an experimental first-line TKI does not seem to reduce imatinib effectiveness in the second line [83,84]. Sequencing TKIs is an important challenge in the management of GIST, and a tailored approach based on an identified resistance mechanism will be an important part of future therapies.

## 6. Conclusions

GIST have a rich molecular landscape that is being unraveled thanks to modern genomic analyses. Four lines of therapy are currently standard, but the prognosis after the second line is poor and optimal management of secondary mutations remains a challenge.

Repeating biopsies to tailor treatments might be a step in the right direction, and liquid biopsies at progression may offer a non-invasive alternative.

New molecules with wider KIT inhibition are being tested and could change the catalog and the sequence of existing treatments. Combination therapies may also be an approach to overcome current resistance mechanisms [85], either by targeting these directly or avoiding their development.

**Author Contributions:** Conceptualization H.M. and A.F.; methodology, H.M. and A.F.; validation, H.M., A.F. and M.M.; formal analysis, H.M., A.F. and M.M.; resources, H.M., A.F. and M.M.; writing—original draft preparation, H.M.; writing—review and editing, H.M.; supervision, A.F. and M.M. All authors have read and agreed to the published version of the manuscript.

**Funding:** This research received no external funding.

**Conflicts of Interest:** The authors declare no conflict of interest.

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
