# Peer review of "Precision Oncology in Gastrointestinal Stromal Tumors"

_curroncol, doi:10.3390/curroncol30050351_

Round 1
Reviewer 1 Report
This review manuscript summarized the sequential therapy of gastrointestinal stromal tumors, GIST from 1st line to late-line or ongoing studies. This comprehensive review based on precision oncology is well-written and clearly presented, which advances the field of this rare cancer.
I have some suggestions in order to improve the manuscript.
1. In Figure 1, they need to show the TK1 domain at exons 13 and 14 according to lines 56-58. Please modify the figure.
2. Regarding imatinib therapy, they stated that the standard dosage is 400mg for all sensitive mutations in Lines 110-111. They need to display more details for the sensitivity of each KIT mutation (exon 11/13/14/17/18) to better understand the biology.
3. Very recently, the HSP90 inhibitor, pimitespib has been approved in Japan and is under clinical trial in Europe and North America. They should add this drug.
CHAPTER-GIST-301 Ann Oncol. 2022 Sep;33(9):959-967.
4. Typo: Line267 TNO55→TNO155
Author Response
All the suggestions have been addressed
Reviewer 2 Report
This review provides a comprehensive summary of current knowledge of diagnosis and therapy of gastrointestinal stromal tumors (GIST).
The manuscript is well structured and written.
I have few comments:
Page 2, line 79: It is not correct to consider as wild-type the GISTs with low frequency KIT or PDGFRA mutations. These must be classified as KIT or PDGFRA mutated.
Page 2, line 83: The single reference provided is not sufficient to support the data reported in this sentence. Add other citations.
Page 3, lines 92-96: these classifications are known also as AFIP classification and modified NIH classification. Include this information.
Page 5, line 180: the NTRK fusions value (5%) of is too high. This is less than 1%.
Reference 70 is not a valid citation. Delete this reference and instead cite the corresponding clinical trial ID.
Author Response
- We rephrased this part of the manuscript for more clarity:
- Prior to the era of modern and extensive sequencing, some wild-type GIST had been falsely classified as such, while actually containing low-frequency KIT or PDGFRA mutations. T
- Today, with modern techniques, they would no longer be considered as wild type.
- All the other suggestions have been addressed
Reviewer 3 Report
In the review titled “Precision oncology in gastrointerstinal stromal tumors”, the authors discussed the background and molecular pathogenesis of GIST and highlighted the treatment management based on the genetic alterations.
There are some concerns that the authors need to address before the publication of this article.
1. There are other similar review articles on this topic. And the discussion in this manuscript is rather general and lacking depth. There should be more new input and knowledge that the authors add
Mei L, Smith SC, Faber AC, Trent J, Grossman SR, Stratakis CA, Boikos SA. Gastrointestinal Stromal Tumors: The GIST of Precision Medicine. Trends Cancer. 2018 Jan;4(1):74-91
Florou V, Trent JC, Wilky BA. Precision medicine in gastrointestinal stromal tumors. Discov Med. 2019 Nov-Dec;28(155):267-276.
Dermawan JK, Rubin BP. Molecular Pathogenesis of Gastrointestinal Stromal Tumor: A Paradigm for Personalized Medicine. Annu Rev Pathol. 2022 Jan 24;17:323-344
2. It would be much easier for readers to understand if a mechanistic figure or table is added to summarize the genetic mutations and the treatment options
Author Response
- As suggested, a table was added to summarize the treatment options according to the underlying primary (and secondary) mutations
For the discussion, to stay concise, we opted to not repeat or summarise what has been said earlier in the text.
Reviewer 4 Report
This is an interesting and informative review on the management of GIST. Although it is generally well-written, I have some suggestions in order to improve the paper and present the information in a more academic way.
1. The punctuation goes after the reference! For example (2).
2. line 39: "GIST occur nearly always in the abdomen" it should be replaced with "GI tract" not the abdomen. For example, the gallbladder is also within the abdomen.
3. Figure 1 caption: "The first line treatment" is very vague. Please clearly indicate the drugs that are illustrated in the figure.
4. Line 132: "Compliance issues are also a major factor to consider when suspecting pseudo-progression" Please explain
5. The subheading of section 3 are too many. For example, in the Non-SDH deficient you may not use subheading. They are more confusing. Describe all the cases together, in different paragraphs, without titles.
6. In second-line, third-line, fourth-line, the subheadings should not include "after failure of...". This is something that should be mentioned in the text, not the title.
Author Response
All the other suggestions have been addressed
Reviewer 5 Report
There is a content overlap in future directions, discussion and conclusions. Please remove that.
Author Response
The suggestion has been addressed
Round 2
Reviewer 1 Report
The authors have addressed all my concerns.
There is just a minor point. Please delete lines 126-128 as this statement is overlapped with lines 115-118.